# Economic Evaluations of Magnetic Resonance Image-Guided Radiotherapy (MRIgRT): A Systematic Review

**DOI:** 10.3390/ijerph191710800

**Published:** 2022-08-30

**Authors:** Alessandra Castelluccia, Pierpaolo Mincarone, Maria Rosaria Tumolo, Saverio Sabina, Riccardo Colella, Antonella Bodini, Francesco Tramacere, Maurizio Portaluri, Carlo Giacomo Leo

**Affiliations:** 1Radiation Oncology, Department of Radiotherapy, Hospital “A. Perrino”, ASL Brindisi, 72100 Brindisi, Italy; 2Institute for Research on Population and Social Policies, National Research Council, 72100 Brindisi, Italy; 3MOVE-Mentis s.r.l., 47522 Cesena, Italy; 4Department of Biological and Environmental Sciences and Technology, University of Salento, 73100 Lecce, Italy; 5Institute of Clinical Physiology, National Research Council, 73100 Lecce, Italy; 6Department of Engineering for Innovation, University of Salento, 73100 Lecce, Italy; 7Institute for Applied Mathematics and Information Technologies “E. Magenes”, National Research Council, 20133 Milan, Italy

**Keywords:** economic evaluation, image-guided radiotherapy, adaptive radiation therapy, MRI-guided radiotherapy, systematic review

## Abstract

Objectives: This review systematically summarizes the evidence on the economic impact of magnetic resonance image-guided RT (MRIgRT). Methods: We systematically searched INAHTA, MEDLINE, and Scopus up to March 2022 to retrieve health economic studies. Relevant data were extracted on study type, model inputs, modeling methods and economic results. Results: Five studies were included. Two studies performed a full economic assessment to compare the cost-effectiveness of MRIgRT with other forms of image-guided radiation therapy. One study performed a cost minimization analysis and two studies performed an activity-based costing, all comparing MRIgRT with X-ray computed tomography image-guided radiation therapy (CTIgRT). Prostate cancer was the target condition in four studies and hepatocellular carcinoma in one. Considering the studies with a full economic assessment, MR-guided stereotactic body radiation therapy was found to be cost effective with respect to CTIgRT or conventional or moderate hypofractionated RT, even with a low reduction in toxicity. Conversely, a greater reduction in toxicity is required to compete with extreme hypofractionated RT without MR guidance. Conclusions: This review highlights the great potential of MRIgRT but also the need for further evidence, especially for late toxicity, whose reduction is expected to be the real added value of this technology.

## 1. Introduction

Worldwide, an estimated 19.3 million new cases of cancer were diagnosed in 2020, with an expected increase to 28.4 million in 2040 [1], and around one-half of all these patients will receive radiation therapy (RT) at some point in their illness [2]. The main challenge of RT is the delicate balance between the radiation dose–response relationship for killing tumor cells and the probability of normal tissue toxicity due to exposure of areas surrounding the target site [3]. For this reason, the adoption of an image-guided approach to direct the beam to the right target is considered to be the most convincing success story of radiation oncology in recent decades [4]. This is currently possible with a broad range of modalities. The current standard for image guidance, X-ray computed tomography (CT), has some limitations: poor image quality in regions with consistent internal motion, caused by respiration and gas, which introduce blurring of soft tissue interfaces [4], and, in particular, the inability to adapt the treatment plan on-line [5,6]. Moreover, CT image-guided RT (CTIgRT) has some safety issues due to X-rays and some degree of invasiveness for the surgical implantation of fiducials [7,8]. Magnetic resonance imaging (MRI)-guided RT (MRIgRT), combines MRI technology and linear accelerators to enable adaptive radiotherapy and marker-less cine imaging during treatment with near real-time target tracking. The combination of more precise target localization with dynamic target information can reduce the impact of motion during radiotherapy delivery, resulting in reduced planning target volume margins [9]. Current areas of application include prostate tumors, oligometastatic disease, pancreatic tumors, central lung tumors, brain tumors, and rectal tumors [9]. Due to better image quality, superior soft tissue contrast, and the non-secondary advantage of the absence of ionization, it represents the first on-line adaptive treatment to be delivered [9,10]. By comparison, MRIgRT requires significant infrastructure and capital costs for technology acquisition and staff training, interdisciplinary skills, and a significantly longer time than conventional external beam RT (EBRT), i.e., with other types of image guidance, including CTIgRT [9]. In this systematic review, we analyze the current evidence regarding the economic impact of MRIgRT to understand whether its adoption, as an alternative to other non-invasive modalities of RT, can be justified. We also provide suggestions for future economic evaluations of this technology.

## 2. Materials and Methods

Preferred Reporting Items for Systematic Reviews and Meta-Analyses (PRISMA) guidelines [11] and ISPOR suggestions for the critical appraisal of systematic reviews with costs and cost-effectiveness outcomes [12] were followed to conduct this systematic review. The literature search is reported according to PRISMA-S (search extension) [13].

### 2.1. Eligibility Criteria

Model-based and empirical health economic studies, and full and partial economic evaluations comparing MRIgRT with other types of external RT, were included. The search was not limited by the geographic setting, the source of funding source, or the time horizon. Studies were excluded if they only considered a health benefit outcome, such as disability-adjusted life years (DALYs) or quality-adjusted life years (QALYs), without costs. Literature reviews and abstracts of conference proceedings were also excluded.

### 2.2. Information Sources

We searched the following electronic databases up to March 2022: MEDLINE(R) and Epub Ahead of Print, In-Process, In-Data-Review & Other Non-Indexed Citations, Daily and Versions(R), through the OVID platform, Scopus, and the international Health Technology Assessment (HTA) database of the International Network of Agencies for Health Technology Assessment (INAHTA). Adopting a “snowballing” approach, we manually screened the reference list of included articles and conducted a systematic citation tracking in Web of Science, Scopus, PubMed, and Google Scholar.

### 2.3. Search Strategy

The search strings were developed by two authors with competences in health technology assessment (CGL and PM) and verified by two authors with expertise in MRIgRT (AC and MP).

We used search terms to identify the technology of interest and the presence of an economic analysis. The NLM-controlled vocabulary thesaurus (MeSH) was adopted and entry terms and synonyms were used to increase search sensitivity. We limited the search to the English language. The complete search strategy can be consulted in the Appendix A. Duplicates were removed by PM using an automatic check with Microsoft Excel based on PubMed ID and DOI.

### 2.4. Selection Process

Search results were retrieved from the databases and double screened independently by all the authors. The first screening was based on the title and abstract and was performed with the support of Abstrackr [14]. Potentially relevant articles were retrieved for full-text review and their eligibility was determined by all the authors. Disagreements were resolved through discussion until consensus was reached.

### 2.5. Data Collection Process

Each selected study was independently evaluated by two authors (CGL and PM) to extract relevant data. In addition, in this case, disagreements were resolved through discussion among all the authors until consensus was reached.

### 2.6. Data Items

Three different types of information were extracted from selected studies: (i) general study characteristics in terms of population, intervention, comparator and study design; (ii) a methodological description of the economic evaluation; and (iii) the results of the analysis. In addition to reporting the objectives of the study and the target audience, the first set of data consists of a description of the characteristics of the patients including their pathological condition, the technologies examined in comparison, the type of model, and the analysis performed. The description of the methodological approach considered the time horizon, the annual discount rate, the costs and health outcomes taken into account and the method for their calculation/identification, the management of uncertainty, the model validation, and the presence of conflicts of interest. With respect to the results of the economic analysis, the following data were collected: the costs and health outcome for all the considered strategies, and other considerations in terms of comparison of the strategies provided by the authors, any limitations declared by the authors, and the conclusions.

The above reported data were collected schematically through specially defined forms to collect sufficient and unambiguous data that faithfully represent the source in a structured and organized manner. In case of missing relevant data in the retrieved works, the authors were contacted. The data collected were then reported in three tables.

## 3. Results

As shown in Figure 1, a total of 969 studies were identified with 234 duplicates that were removed. Of the 735 remaining publications, 728 records were excluded on the basis of abstracts and titles for the following reasons: MRIgRT not present, economic evaluation not performed, or abstracts of conference proceedings, reviews, or non-original research (e.g., letters or commentaries). One of the seven full studies retrieved after title and abstract screening was not found even after writing to the authors [15]. Another [16] was excluded after full-text screening due to the absence of a comparator. Five studies were therefore included in this review (see Table 1).

### 3.1. Study Type

Of the five studies, two were empirical-based and adopted a time-driven activity-based costing [17,18], whereas three were model-based and performed either a cost minimization analysis [19] or a cost-utility analysis (CUA) [20,21]. Three studies were based in the United States [17,18,20], one in Australia [19], and one in the Netherlands [21], although the last two were based on healthcare processes formalized in the USA context. One study [17] focused on subjects with unresectable hepatocellular carcinoma (HCC), whereas all the others focused on patients with prostate cancer (PCa). All the studies assessed the five-fraction MRIgRT, and Schumacher et al. [20] also considered a 39-fraction program. Four studies focused on the 0.35 and one [21] on the 1.5 Tesla version for the MRI module although, in this case, despite the declared intentions of the authors, only data from the 0.35 Tesla version were adopted in the analysis. No study considered all the possible alternatives (i.e., with and without image guidance) to MRIgRT: four focused only on CTIgRT and one [21] on a generic EBRT without details on the typology of image guidance (from now on, we refer to this simply by EBRT when dealing with Hehakaya et al. [21]). Hehakaya et al. [21] also performed a comparison with low-dose-rate brachytherapy, which was not considered in this systematic review focusing only on non-invasive modalities of RT. An option with extreme hypofractionation (a five-fraction regimen) was considered in all the studies; hypofractionation (20 fractions) was also assessed in Hehakaya et al. [21] and conventional fractionation (39 fractions) by Schumacher and colleagues [20] and Hehakaya and colleagues [21].

### 3.2. Adopted Methodologies

A summary of the methodologies adopted is reported in Table 2.

As shown in Figure 2, the studies are closely interrelated and therefore the numerous similarities are not surprising. The most relevant difference between Parikh et al., 2020 [17] and Parikh et al., 2021 [18] is the population: subjects with localized unresectable HCC and with localized PCa eligible for stereotactic body radiation therapy (SBRT), respectively. These studies share half of the authors and follow the same methodology with some minor differences. All the costs included in Hehakaya et al. [21] for the MRIgRT strategy, except those of complications and recurrence, were also adopted by Schumacher et al. [20]. Berber et al. [19] adapted the US-based estimates of all the costs considered from Parikh et al., 2020 [17] and Schumacher et al. [20] at the Australian hourly wage levels.

### 3.3. Model Structure

The two studies performing the CUA, Schumacher et al. [20] and Hehakaya et al. [21], specified the time horizon and discount rate adopted (see Table 2). Different choices were made. In particular, in Hehakaya et al. [21], distinct discount rates were applied to costs and benefits, and the same rates were adopted in Schumacher et al. [20]. The choice of Hehakaya and colleagues [21] was consistent with the Dutch Guidelines for Economic Evaluations in Healthcare [22] to account for the growing value of health benefits in the future. In Berber et al. [19], these parameters were declared not applicable even though the authors considered the costs of late toxicity.

In all studies, the health care provider perspective was adopted and, consequently, only the costs incurred by the provider were considered. Schumacher and colleagues [20] pointed out that, being the study focused on prostate RT patients and therefore dealing mostly with retirees who do not contribute to the workforce, a broader societal perspective was unlikely to have changed the results significantly. This consideration could be applied to all the other studies on PCa, even if it neglects lost working days of caregivers. Only the Dutch study [21] considers direct non-medical costs (travel expenses) but, in this State, these costs are partially reimbursed by health insurance when needed for cancer treatment, so they do not produce a change in the perspective adopted.

Direct medical costs were considered with some differences among studies. All the authors included acquisition, installation, and maintenance. Quality assurance was considered in all the studies except Schumacher et al. [20] and Hehakaya et al. [21]. Space was accounted for in Parikh et al., 2020 [17] and Parikh et al., 2021 [18], and training and accreditation only in Berber et al. [19]. Personnel costs were derived from reference hourly wages and interviews with staff in Parikh et al., 2020 [17], Parikh et al., 2021 [18], and Schumacher et al. [20], and, in one case [18], operation times were complemented by direct observation of the RT procedures. As reported above, Berber and colleagues [19] and Hehakaya and colleagues [21] assumed the operational costs from Parikh et al., 2020 [17] and/or Schumacher et al. [20]. In Berber et al. [19], Schumacher et al. [20], and Hehakaya et al. [21], the cost of RT complications was considered and it was derived from an independent review of the literature. All these studies included acute and late gastrointestinal (GI) and genitourinary (GU) side effects. Biochemical recurrence was also taken into account by Schumacher and colleagues [20] and Hehakaya and colleagues [21]. The two studies that performed a CUA [20,21] assigned utilities that were found in literature and had only one work in common: Stewart et al. [23]. Berber et al. [19] and Hehakaya et al. [21] are the only studies that adopted GI/GU toxicity rates specifically related to the application of MRIgRT: respectively, Alongi et al. [24], a preliminary report from a prospective observational study for the clinical use of 1.5 T Elekta Unity on the feasibility, quality of life, and patient-reported outcome measures for localized PCa treated with SBRT, and Bruynzeel et al. [25], a prospective single-arm phase 2 study of RT with 0.35 T MRIdian (ViewRay Inc.) for PCa. It is worth noting that Berber and colleagues adopted the evidence from a study on a 1.5 T MRIgRT although they focused on the impact of a 0.35 T model.

The treatment process (administration of RT) was assessed in detail in Parikh et al., 2020 [17], Parikh et al., 2021 [18], and Schumacher et al. [20], whereas it was assumed from previous works [17,20] in Berber et al. [19] for recalculating costs only. Hehakaya and colleagues [21] did not focus on the process, having directly assumed its cost from Schumacher et al. [20].

A Markov model was explicitly adopted in two studies [20,21] for post-treatment states. The cycle length was set to 1 year in both.

The structure of the state transition model, in terms of how different health states are classified and how subjects move among them, was provided by Hehakaya and colleagues [21] and, in a simplified version, by Schumacher and colleagues [20]. Both the works, aside from the details on the severity of complications that were not addressed in the latter, adopted the same health states. In Schumacher et al. [20], the transition probabilities of the simulated patients undergoing MRIgRT were derived from those adopted for CTIgRT and changed by applying a 1% reduction. In Hehakaya et al. [21], the base case assumed specific transition probabilities for the MRIgRT strategy that were taken from a population undergoing MRIgRT. No study formally provided a transition matrix, but studies only used absolute annual probabilities of early and late complications. In Schumacher et al. [20], graphical representations of health state transitions were provided in a simplified version: the severity of complications—Grade 2 vs. Grade ≥ 3—although included in the model, was not explicitly shown. Transition probabilities were also retrieved from the literature. In Schumacher et al. [20], a systematic search in PubMed was conducted to identify the studies reporting the outcomes of daily CTIgRT and the annual toxicity rates were calculated by dividing the average toxicity rates reported by the average number of years of follow-up. In Hehakaya et al. [21], a selection of trials was considered to retrieve urinary and bowel complications in terms of transition probabilities and utilities; in this case, no extrapolation method was declared. Due to the lack of evidence, both works assumed that cancer outcomes for MRIgRT were the same as for CTIgRT [20] or as for conventional EBRT [21].

The two studies that performed a CUA [20,21] assumed that the post-treatment utilities of patients undergoing MRIgRT were similar to those of performing an alternative RT (CTIgRT in Schumacher et al. [20] and conventional EBRT in Hehakaya et al. [21]).

Both Schumacher et al. [20] and Hehakaya et al. [21] aimed to answer the same research question: what are the side-effect reductions needed to make MRIgRT strategy dominant in terms of a target incremental cost-effectiveness ratio (ICER)? Authors adopted different thresholds for ICER: 50,000 and 100,000 USD/QALY in Schumacher et al. [20], and 80,000 EUR/QALY in Hehakaya et al. [21].

### 3.4. Uncertainty

All the studies managed uncertainties through deterministic one-way sensitivity analyses: mean input parameters changed with standard deviation or ± fixed percentages. In some cases, further sensitivity analyses were performed by changing the number of fractions in the RT plan [17,18,19], and, only for Parikh et al., 2020 [18] and Hehakaya et al. [21], by modulating other model parameters such as lifetime of the equipment and fiducial marker placement. Only two [20,21] of the three studies that took utilities into account reported uncertainty in the utility estimates.

Aside from Berber et al. [19], all the other studies reported some conflicts of interest. Healthcare technology companies have granted research funds [17,18,21] or honoraria [17,18], or have covered travel expenses [17,20]. Some of the authors in Parikh et al., 2021 [18] are shareholders in ViewRay Inc. (one of the few companies in the world that produces the MRIgRT technology). Elekta AB (Stockholm, Sweden), the company that produces the 1.5 T Elekta Unity that is used to perform MRIgRT, and Philips Medical Systems (Best, The Netherlands), a company that produces systems for treatment planning, partially funded several MR-Linac scientific projects at the Division of Imaging and Oncology of University Medical Center Utrecht [21]. ViewRay Inc. directly supported a research organization or single authors in Parikh et al., 2020 [17], Parikh et al., 2021 [18], and Schumacher et al. [20]. Varian Inc. (a CTIgRT machine manufacturer) directly supported some of the authors of Parikh et al., 2020 [17] and Parikh et al., 2021 [18] through consulting fees, travel expenses, and/or advisory positions. Three studies, Schumacher et al. [20], Berber et al. [19], and Hehakaya et al. [21], were sponsored by public institutions, and Parikh et al., 2021 [18] received a grant from ViewRay. No information regarding funds was reported in Parikh et al., 2020 [17].

### 3.5. Model Validation

Only Parikh et al., 2020 [17] and Hehakaya et al. [21] reported a model validation phase in which the authors described what was validated and by whom, but details on the methodology adopted were not shown. In the former case, the mapping of the process flow, including the time for activities, was validated on the basis of the input from health personnel (nurses, dosimetrists, physicists, attending physicians from radiation oncology and interventional radiology, front office personnel, and radiation therapists). In addition, treatment times were further validated with patient-level data (no details were provided on this). In the latter case, model assumptions and structure, and input parameters, were discussed with methodological and clinical experts (no details provided, not even on the cross-validation of the model nor for the request to use the model addressed to an independent expert).

### 3.6. Model Outcomes

The results of the economic analysis are shown in Table 3.

For the sake of comparability, in Table 3, we also provide all the costs related to the prices of year 2022 and converted into USD, taking into account purchasing power parities between countries. This was undertaken using the web-based tool developed by the Campbell and Cochrane Economics Methods Group and the Evidence for Policy and Practice Information and Coordinating Centre [27], as explained in Shemilt et al. [28].

Three studies provided details on the capital and operational costs for performing RT, which have been grouped into: (1) space, equipment, and maintenance, (2) personnel, and (3) materials (Parikh et al., 2020 [17], Schumacher et al. [20], and Berber et al. [19], although the last merged the second and third cost categories). Some authors reported the cost for purchasing the equipment: 7,800,000 2020 USD (8,127,000 2022 USD) [17] and 9,000,000 2019 USD (9,573,000 2022 USD) [20], in both cases for a 0.35 T MRIgRT.

In the MRIgRT strategy, costs for equipment and maintenance are the most relevant costs (apart from the strategy with five fractions in Schumacher et al. [20]) and their amount is greater than that of the technologies compared. For all the other strategies, personnel and material have the greatest magnitude. The two studies of Parikh found CTIgRT was superior for HCC [17] and PCa [18] but this result could have been highly conditioned by the absence of an appraisal of acute and late toxicity, which is expected to be the real strength in favor of MRIgRT. Berber et and colleagues [19], although the results of the cost minimization analysis are favorable to CTIgRT, recommend listing the MRIgRT on the Medical Benefits Schedule. Their cost analysis could have been biased by the lack of evidence regarding the treatment complications. Schumacher and colleagues [20] and Hehakaya and colleagues [21] managed this limitation in the body of evidence: they assumed a percentage degree of toxicity as a variable parameter in their model that was set (as a target point) to reach cost efficacy. Berber and colleagues [19] included in their model all the possible grades of acute and late GI/GU toxicity for CTIgRT while considering only grade 1 and 2 acute GI/GU toxicity for MRIgRT. This may have led to an underestimation of adverse effects for the MRIgRT strategy that can explain the large reported difference in the cost of toxicity between the modeled strategies: AUD 150.63 vs. AUD 1592.95, for MR and CT, respectively. The two studies that performed a CUA [20,21] showed a very wide range of percentage reductions in toxicity for MRIgRT to be cost effective compared to the alternative: from 0% to 94%, depending on the scenario considered. Notably, for Schumacher and colleagues [20], the side-effect reduction thresholds for cost-effectiveness of 39-f MRIgRT compared with 39-f CTIgRT were 94 and 50% using standard ICER ratios of 50,000 and 100,000 USD/QALY, respectively. For 5f MRIgRT, the side-effect reduction thresholds were 14 and 7%, respectively, when compared with 5f CTIgRT. In Hehakaya et al. [21], based on an ICER of 80,000 EUR/QALY, the 5f-MRIgRT becomes cost effective compared with a 5f-EBRT when complications are reduced by 54%. Compared to 20- and 39-fraction treatments, the 5f-MRIgRT was always found to be cost-effective, even with the same level of toxicity.

### 3.7. Quality Assessment

A quality assessment was originally planned as part of this study. Having piloted various tools [29,30], it became evident that they can lead to misleading findings. In fact, the tools were designed for cost-effectiveness studies with both costs and outcome data mainly from trials as the default. Our review, however, contains both costing [17,18,19] and cost-effectiveness studies [20,21], and the former “scored” consistently lower as they do not cover all the domains. Furthermore, it should be considered that, due to the absence of a consistent body of evidence on the efficacy of MRI in reducing the toxicity of RT, all the included studies have major limitations and, as stated by the authors, should be considered as preliminary assessments to provide indications in the current initial phase of application of the technology. We therefore chose not to undertake a formal quality assessment but to adopt a narrative approach by commenting on methodological choices.

## 4. Discussion

### 4.1. Main Considerations

A total of five studies were identified, all published in 2020 or 2021. Although the methodologies adopted were different (two performed a full economic assessment, one a cost minimization analysis, and two a time-driven activity-based costing), all authors had a similar purpose: to provide healthcare systems and governments with a synthesis of the scarce available evidence and insights into the resulting uncertainty. This need has grown in importance over the past two decades as new technologies were developed at a dizzying pace [31]. Indeed, health economics modeling has started to be used even in the early stages of technology development, hence referred to as “early HTA”, when both costs and effects of the innovation are still largely unknown [32,33]. In our opinion, all the included studies could be considered an example of early HTA. In fact, all the authors identified the expected limits in the available evidence and found a way to address them and come to a conclusion. Two approaches were adopted. Three studies [17,18,19] only considered major cost drivers (initial investments and a detailed description of operation costs [17,18] and complications [19]). As is known, this is only one part of the picture. As reported by Sorenson and colleagues [34], to better understand the relation between innovation and spending, it is important to consider the circumstances through which an investment may lead to higher values. Only two studies [20,21] went in this direction since their economic model also included benefits, in the form of expected reduced toxicity in PCa patients. In this case, authors managed the scarce evidence as a model parameter that varied to achieve a predetermined economic target. Consequently, these preliminary analyses, while moving away from a traditional use of health economics modeling and despite the scarce available evidence, provide useful inputs to guide decisions, just as required by early HTA [32]. In the two works by Parikh’s group [17,18], MRIgRT showed higher costs than CTIgRT but the authors acknowledged that their result is sensitive to various assumptions and to individual patterns of practice. Berber and colleagues [19] concluded their HTA report in favor of the adoption of MRIgRT. In the two CUAs, MRIgRT reached cost efficacy, even with a low reduction in toxicity (less than 15%) when extremely hypofractionated and compared with extremely hypofractionated CTIgRT [20], or with conventional or moderately hypofractionated EBRT [21]. Conversely, greater reduction in toxicity was required when conventionally fractionated [20] or to compete with extremely hypofractionated RT [21]. It is worth noting that Hehakaya and colleagues [21] assumed the same cost per fraction (EUR 233, 2007 price level) for all the different compared scenarios, without taking into account the additional costs, in terms of equipment and personnel, needed to ensure the level of accuracy required to provide higher doses in hypofractionations.

One study focused on HCC [17], but the authors did not consider data on efficacy and safety. PCa was the target condition for the remaining four studies. The advantage of using an adaptive approach to this type of treatment is controversial [35]. On the contrary, the peculiarity of MRIgRT is the opportunity for great control of inter- and intra-fraction organ motion, especially for targets deeply influenced by respiratory and cardiac cycle. Treatment of lung and hepatic lesions could result in even greater advantages and deserves attention [36].

Specific considerations can be applied to data sources in Hehakaya et al. [21] and Berber et al. [19]. In the former work, the choice to assess the 1.5 T Elekta AB’s Elekta Unity while adopting the evidence relating to ViewRay Inc.’s 0.35 T MRIdian is not appropriate. The two technologies adopt a different magnetic field with a potential impact on the safety profile. A higher magnitude, in fact, is known to increase the electron-return effect with profound changes in dose distributions near air–tissue interfaces [37]. This leads to unwanted doses deposited on protruding anatomic structures by electrons swept away from the treatment area [38]. Additionally, MRIdian is the only technology that adopts automated beam gating for precise and accurate dosing in order to promptly stop the beam as the tumor moves. Similarly, Berber and colleagues [19], who focused on the 0.35 T MRIdian, used the evidence from a study with 1.5 T Elekta Unity to estimate the probability of occurrence of RT complications.

In approaching the long-term effects of a treatment, another important element is how these may be actualized to allow for a comparison among different technologies.

Only three works considered effects extending over more than 1 year [19,20,21]. They adopted a different approach to report the present value of future costs and health outcomes in their models, indeed reflecting the uncertainty around the choice of the most appropriate value for discount rates and the lack of consensus among economists on the topic [39]. Discount rates play a crucial role in economic evaluation studies and can change resource allocation and prioritization of healthcare [40]. Discount rates, in fact, change the weight of future events (the lower the rates, the higher the weights). The included studies that considered the long-term impact of the adopted technology clearly showed that MRIgRT is characterized by greater initial costs (for investments and for treatment execution) but by presumed less future negative events (complications and recurrence). Consequently, the lower the discount rate for benefits and costs, the greater the advantage for MRIgRT. Berber et al. [19] considered late toxicity but without formally applying a discount rate, hence assuming, as a matter of fact, the lowest option as possible. Schumacher and colleagues [20] chose a 3% discount rate for both costs and benefits, an approach widely adopted in transnational HTA guidelines [41,42]. A differential discounting (4% for costs and 1.5% for effects) was considered by Hehakaya and colleagues [21] who, being based in the Netherlands, literally followed the recommendation of the Dutch Guidelines for Economic Evaluations in Healthcare [22] on this point. As late events are temporarily coupled with related costs (for medical expenses to treat toxicities and recurrence), we believe there is no relevant impact in defining different discount rates for costs and effects in this case, outside the difficulty in comparing different studies (e.g., in meta-analyses).

### 4.2. Recommendations for Future Cost-Effectiveness Models

Any future model should take into account the entire diagnostic pathway of the patients for whom an MRIgRT is preferable. For example, it is important to consider that not all the patients can undergo an MRI examination for several reasons [9]: presence of some types of implanted metal, electronic devices, significant claustrophobia, or bodily impediments. Similar restrictions can apply to CTIgRT as it requires the implantation of fiducial markers. Not all patients can undergo this invasive procedure, due to comorbidities and drugs such as anticoagulants, or because of not accepting the risk of complications resulting from marker placement, such as bleeding, pneumothorax, and infections [43,44]. Introducing this issue in a model results in a reduction in the number of treatable patients, which is a critical element for technologies with high capital and maintenance costs (higher per-patient cost). Alternative paths that consider different fractionations should account for the differences: (a) in the equipment needed in order to ensure the greater accuracy required for higher doses; and (b) in the treatment time of RT sessions, which depends on the delivered dose for the fraction and on the treatment workflow [45]. Time for each activity during CTIgRT is related to the original RT plan and does not vary for each fraction. Instead, the MRIgRT workflow can be deeply affected by daily on-line adaptions, especially with hypofractionation and narrow margins. These refinements allow a better approximation of the number of patients who can be treated in the modeled period. The lack of evidence imposes great attention to the source of data. In addition to what has been previously reported, it is important, where possible, to select the evidence on utilities also considering the country, the setting, and the evaluation technique adopted in the deriving studies. Using review studies, such as Torvinen et al. [46] for the topic of this review, may help in the choice.

In a relatively novel area of application such as that of MRIgRT, the management of uncertainty is critical. A probabilistic sensitivity analysis, instead of a deterministic approach, is a widely recognized means of adding value [29,47].

### 4.3. Strengths and Limitations of Review

The strength of this systematic review is that it is the first that provides an overview of the economic impact of the innovation introduced by MRIgRT, on which high expectations are placed for a positive impact on cancer patients. The reported limitations allowed us to derive suggestions for the improvement of future research.

The studies are too heterogenous for a direct comparison of the cost-effectiveness results.

## 5. Conclusions

In the global economic situation, characterized by a high risk of stagflation, it is ever more important to ensure that technologies ensure good value for money. The studies identified in this review highlighted the potential improvement in health outcomes deriving from the introduction of MRIgRT (reduction in complications and recurrences and a decrease or, alternatively, an acceptable increase, in costs).

In consideration of the detected methodological problems, we propose some recommendations for future studies. To ensure that the cost effectiveness of MRIgRT is robustly assessed, it is necessary to take into account the entire diagnostic path, model the treatment time as a function of fractionation, consider QoL estimates to be as consistent as possible with the modeled scenario, and take full account of the uncertainty.

Bearing in mind the potential impact on a large number of cancer patients, further cost-effectiveness analyses are therefore strongly recommended as new evidence from on-going prospective trials [9] becomes available.

## Figures and Tables

**Figure 1 ijerph-19-10800-f001:**
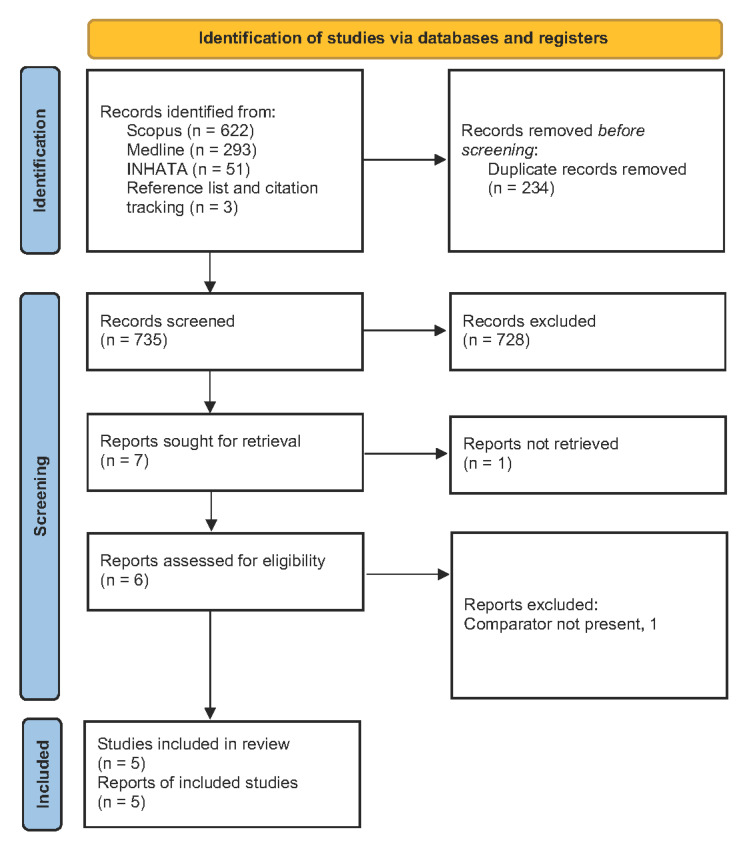
PRISMA 2020 flow diagram [11].

**Figure 2 ijerph-19-10800-f002:**
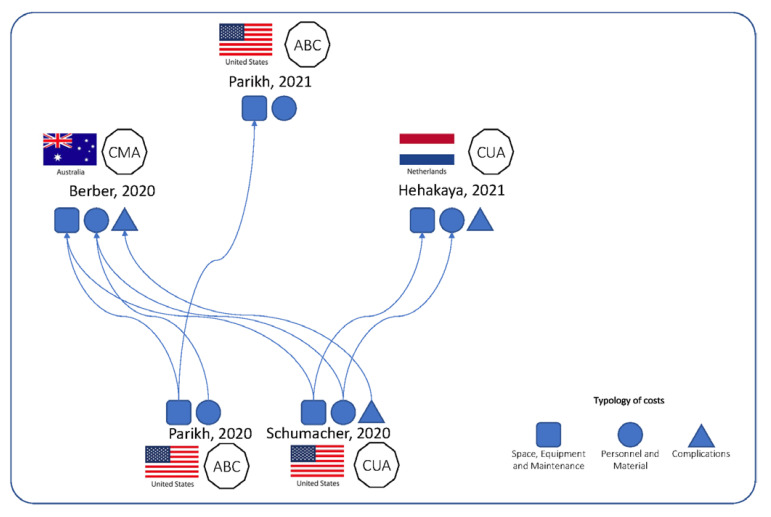
Graphical description of interrelations among studies [18,19,20,21].

**Table 1 ijerph-19-10800-t001:** Characteristics of the included studies.

Study	Aim	Target Audience	Type of Economic Analysis	Country	Population	Compared Technologies		Study Design
						MRIgRT	Other RT Modalities	
Parikh, 2020 [17]	To determine and compare the direct cost of treatment of SBRT using CTIgRT and MRIgRT	Decision makers for investment choices	TDABC	California, USA	Subjects with localized unresectable HCC	0.35 T 5f-MRIgRT SBRT	5f-CTIgRT SBRT	Empirical (expert opinion through interviews)
Parikh, 2021 [18]	To determine the difference of direct cost of treatment of SBRT using CTIgRT and MRIgRT	Decision makers for investment choices	TDABC	California, USA	Subjects with localized PCa eligible for SBRT	0.35 T 5f-MRIgRT SBRT	5f-CTIgRT SBRT	Empirical (interviews/surveys with departmental personnel); CTgRT and MRgRT treatment times measured from local patients undergoing prostate SBRT
Berber, 2020 [19]	To determine the costs of two compared image-guided radiotherapies (MR and CT)	Decision makers for investment choices	CMA	Australia	Subjects with PCa undergoing EBRT	0.35 T 5f-MRIgRT SBRT	5f-CTIgRT SBRT	Model-based
Schumacher, 2020 [20]	To determine the toxicity reduction required to justify the added costs of MRIgRT over CTIgRT for the treatment of localized prostate cancer	Decision makers for investment choices	CUA	Florida, USA	Subjects with PCa (median age of prostate cancer diagnosis is 66 years old)	- 0.35 T 5f-MRIgRT SBRT; - 0.35 T 39f-MRIgRT	- 5f-CTIgRT SBRT; - 39f-CTIgRT	Model-based (Markov model). TDABC to determine the costs. Threshold analysis for CUA
Hehakaya, 2021 [21]	To estimate the relative minimally required reduction in grade ≥2 urinary, grade ≥2 bowel, and sexual complications in patients with low- and intermediate-risk localized PCa, and the maximum price of 5-fraction MRIgRT to be cost-effective, compared to current radiotherapy regimens	Decision makers for investment choices; researchers for future studies on prospective cost- effectiveness analysis	CUA	The Netherlands	Hypothetical cohort of 1000 men with low-/intermediate-risk localized PCa and no other severe comorbidities, treated at age 65 years	1.5 T 5f-MRIgRT SBRT	5f-, 20f- or 39f-EBRT	Model-based (state transition model, threshold analysis)

**Table 2 ijerph-19-10800-t002:** Methodologies adopted in the studies.

Study	Time Horizon	Annual Discount Rate	Type of Costs			Sources for Calculation of Costs	Health Outcomes	Sources for Calculation of Health Outcomes	Treatment of Uncertainty	Model Validation	Conflicts of Interest and Sources of Funding
			Direct Medical	Direct Non-Medical	Information Presented in Natural Units						
Parikh, 2020 [17]	N.A.	N.A.	(1) Space, Equipment and Maintenance; (2) Materials; (3) Personnel	No	Details provided in terms of time/patient in each process phase (new patient, simulation, planning, treatment, on treatment visit, follow-up visit, quality assurance) and for each personnel category (attending interventional radiologist, attending radiation oncologist, dosimetrist, environmental services staff, front desk staff, imaging technologist, medical/hospital assistant, nurse, physicist, radiation therapist, technician, transporter)	Inputs from: - healthcare professionals from the clinical center involved in the empirical study; - sales representatives	N.A.	N.A.	One-way deterministic sensitivity analyses (input parameters changed with ± 20%). Additional sensitivity analyses: 3 and 7 fractions instead of 5 for MRIgRT	Process flow maps and their various subcomponents, including the probability of time spent during the activity and specific resources used by each activity, were formulated and validated on the basis of input from nurses, dosimetrists, physicists, attending physicians from radiation oncology and interventional radiology, front office personnel, and radiation therapists. In addition, treatment times for MRIgRT and CTIgRT were further validated with patient-level data.	Several authors received research support or honoraria from ViewRay or Varian
Parikh, 2021 [18]	N.A.	N.A.	(1) Space, Equipment and Maintenance; (2) Materials; (3) Personnel	No	No detail provided	Inputs from healthcare professionals, department chief financial officer, and sales representatives	N.A.	N.A.	One-way deterministic sensitivity analyses (input parameters changed with ± 20%). Additional sensitivity analyses were performed that involved modifications of the number of fractions (1 and 7 instead of 5) for MRIgRT and CTIgRT	N.R.	Several authors are consultants, employees, or shareholders of ViewRay or Varian. The study received a grant from ViewRay
Berber, 2020 [19]	N.A.	N.A.	(1) Space, Equipment and Maintenance; (2) Materials; (3) Personnel (4) Costs of RT complications (acute and late GI/GU toxicity)	No	Details provided in terms of time/patient in each process phase (patient registration, pre-clinic charting, clinic visit, post-clinic visits, MR clearance process, fiducial marker placement, post-op time (after fiducial placement), simulation, review of images, after simulation, treatment planning, prior treatment, treatment, follow-up visit) and for each personnel category (front desk, radiation oncologist, medical assistant, interventional radiologist, medical physicist, medical assistant, nurse, radiation therapist, dosimetrist, imaging technologist, technician)	[17,20] for cost. Acute and late GI/GU toxicity rates were obtained from a systematic search of literature.	N.A.	N.A.	One-way deterministic sensitivity analyses (input parameters changed by ± 20% and ± 50%). Additional sensitivity analyses were performed that involved modifications of the number of fractions for CTIgRT from 5 fractions to 30 and by removing cost of fiducial marker placement.	N.R.	The authors declared the absence of conflicts of interest. The report was commissioned by the AGDH.
Schumacher, 2020 [20]	15 years	3%	(1) Space, Equipment and Maintenance; (2) Materials; (3) Personnel (4) Costs of RT complications (acute and late GI/GU toxicity) and BCR	No	Details provided in terms of time/patient in each process phase (consultation, simulation, planning, treatment, on treatment visit, follow-up visit) and for each personnel category (physician, nurse, receptionist, dosimetrist, therapist)	The costs of purchasing and maintaining CTIgRT and MRIgRT units were obtained by reviewing the literature. TDABC was used to determine the cost of all steps of patient care. For each step, personnel time and costs were determined using literature values, and interviews with staff and records of staff salaries of the clinical center involved in the empirical study. Costs of complications and BCR were obtained using previous reports in the literature, databases, and Medicare Physician Fee Schedule.	QALYs	A literature search reporting outcome of daily CTIgRT. The base probabilities of toxicities were then reduced by a relative 1% when using MRIgRT. QALYs were obtained using previous reports in the literature and databases	One-way sensitivity analyses were performed on conventional therapy and SBRT using 50,000 and 100,000 USD/QALY. The ranges for all costs were based on literature values or ±25% of the base estimate, except utilities which were ±0.10 of the base estimate.	N.R.	One of the authors received travel funding from ViewRay. The project was partially supported by the NCI-NIH
Hehakaya, 2021 [21]	From 65 years until death	1.5% for utilities; 4% for costs	Comprehensive cost that includes: (1) Space, Equipment and Maintenance; (2) Materials; (3) Personnel (4) Costs of RT complications (acute and late GU, acute and late GI) and BCR	Travel expenses	No detail provided	Derived from published health economic evaluations in radiotherapy, the Dutch guideline for costing research, and the Dutch online database for medication costs	QALYs	MRIgRT utilities assumed as similar to post-treatment utilities as conventional EBRT (from literature).	One-way deterministic sensitivity analyses (mean input parameters changed with standard deviation or ± 20%)	(1) Model structure, input parameters, and discussion of major model assumptions undertaken with methodological and clinical experts; (2) Model performance appraised by using it similarly by an independent expert and by building it with two different software applications; (3) Model cross-validation through a structured literature search to compare model structure, assumptions, and outcomes of interest with cost-utility models.	The authors declare no personal conflicts of interest. Funds from ZonMw

**Table 3 ijerph-19-10800-t003:** Results of economic analyses.

Study	MRIgRT Costs/Patient *		Other RT Modalities Costs/Patient *		Other Considerations	Limitations	Conclusions
	Direct Medical	Direct Non-Medical	Direct Medical	Direct Non-Medical			
Parikh, 2020 [17]	(1) Space, equipment and maintenance: 4769 2020 USD (4969); (2) Personnel: 3603 (3754); (3) Materials: 250 (260); TOTAL: 8622 (8983)	N.C.	(1) Space, equipment and maintenance: 2912 (3034); (2) Personnel: 3752 (3909); (3) Materials: 642 (669); TOTAL: 7306 (7612)	N.C.		Estimates drawn from a single institution’s processes, salary data, and space and equipment. Data obtained from personnel interviews instead of from measured times for specific patient encounters	The estimated direct costs to treat patients who have localized unresectable HCC with MR-guided SBRT are 18% higher than with CT-guided SBRT, although this difference in cost is sensitive to various assumptions and will vary based on individual practice patterns
Parikh, 2021 [18]	N.R.	N.C.	N.R.	N.C.	Differences in costs between MRIgRT and CTIgRT **** (1) Space, equipment and maintenance: 1542 2021 USD (1571); (2) Personnel: 210 (214); (3) Materials: −255 (−260); TOTAL: 1497 (1526)	Estimates of personnel and material costs drawn from a single institution’s analysis. The equipment costs used in the analysis taken from sales representatives. When accounting for different fractionation regimens (e.g., 1 fraction or 7 fractions vs. 5 fractions), the approximate cost per fraction was kept constant and not explicitly accounted for the variable length of treatment time depending on nominal dose delivered	The base case of the analysis estimates USD 1497 in increased direct costs utilized by delivering prostate SBRT with MRIgRT instead of CTIgRT, although modifications to key model inputs may change this result.
Berber, 2020 [19]	(1) Space, equipment and maintenance: 4292.09 2020 AUD (3110); (2–3) Personnel & Material: 1623.33 (1176) (4) Complications: 150.63 (109) TOTAL: 6066.05 ** (4396)	N.C.	(1) Space, equipment and maintenance: 689.5 (500); (2–3) Personnel & Material: 1846.34 (1338); (4) Complications: 1592.95 (1154); TOTAL: 4128.79 *** (2992)	N.C.			The general conclusion of the report was in favor of listing MRIgRT on the Medical Benefits Schedule
Schumacher, 2020 [20]	(A) Conventional 39f-MRIgRT: (1) Space, equipment and maintenance: 12,406 2019 USD (13,196); (2) Personnel: 6225 (6621) (3) Materials: 205 (218) TOTAL (1 + 2 + 3): 18,836 (20,035) (4) Complications not provided on a per patient basis but detailed for feeding the Markov model. (B) 5f-MRIgRT: (1) Space, equipment and maintenance: 2118 (2253); (2) Personnel: 4664 (4961); (3) Materials: 35 (37); TOTAL (1 + 2 + 3): 6816 (7250) (4) Complications as above	N.C.	(A) Conventional 39f-CTIgRT: (1) Space, equipment and maintenance: 2955 (3143); (2) Personnel: 5752 (6118); (3) Materials: 0 (0) TOTAL (1 + 2 + 3): 8707 (9261) (4) Complications not provided on a per-patient basis but detailed for feeding the Markov model. (B) 5f-CTIgRT: (1) Space, equipment and maintenance: 379 (403); (2) Personnel: 4549 (4839); (3) Materials: 430 (457); TOTAL (1 + 2 + 3): 5357 (5698) (4) Complications as above	N.C.	Percentage reduction in complications to reach cost-efficacy: (i) Target ICER 50,000 USD/QALY (A) 39f-MRIgRT: 94%; (B) 5f-CTIgRT: 14%; (ii) Target ICER 100,000 USD/QALY (A) 39f-MRIgRT: 50%; (B) 5f-CTIgRT: 7%	The lack of technology data is the most important limitation	MRI-IGRT can easily be cost-effective for stereotactic prostate cancer treatment as only a slight reduction in overall side-effects is required (7% using 100,000 USD/QALY). Conventional fractionation would require a greater side-effect reduction (50% using 100,000 USD/QALY), but cost-effectiveness remains possible. A randomized clinical trial comparing MR-IGRT to CT-IGRT would better control for variations in the assumptions required to produce this model
Hehakaya, 2021 [21]	(1–3) Space, equipment and maintenance, Personnel and Materials costs all together: 5830 2019 EUR (7789) (4) Complications not provided on a per patient basis but detailed for feeding the Markov model.	842 (630)	(1–3) Space, equipment and maintenance, Personnel and material costs all together: (A) 5f-EBRT: 1165 (1556); (B) 20f-EBRT: 4660 (6225); (C) 39f-EBRT: 9090 (12,144) (4) Complications not provided on a per patient basis but detailed for feeding the Markov model.	(A) 5f-EBRT: 470 (628); (B) 20f-EBRT: 1870 (2498); (C) 39f-EBRT: 3650 (4876)	Percentage reduction in complications of 5f-MRIgRT to reach cost-efficacy (ICER 80,000 EUR/QALY): (A) 5f-EBRT: 54%; (incremental costs USD 6610 (4948 EUR); incremental QALYs 0.06); (B) 20f-EBRT: 0%; (C) 39f-EBRT: 0%	(1) Lack of technology data (for instance, combined health states and post-treatment utility; some data taken from a similar technology 0.35 T MRIgRT instead of the 1.5 T version). (2) Dutch cost data to estimate cost-effectiveness with a consequent limited applicability in other countries.	MRIgRT is found to be cost-effective compared to 20f- and 39f-EBRT with no further reduction in complications. More challenging scenarios exist for 5f-EBRT in which rates of complications or costs need to be reduced significantly to come to cost-effective out- comes. Cost-effectiveness outcomes are highly sensitive to biochemical progression and utilities of urinary, bowel and sexual complications.

* Costs are in reported currency with USD 2022 costs in brackets to aid comparison. ** The authors reported AUD 6056.67. *** The authors reported AUD 4126.29. **** Positive if higher for MRIgRT than CTIgRT. Legend: CTIgRT = computed tomography image-guided radiotherapy; EBRT = external beam radiotherapy; HCC = hepatocellular carcinoma; MRIgRT = magnetic resonance image-guided radiotherapy; N.A. = Not applicable; N.C. = Not calculated; SBRT = stereotactic body radiotherapy; TDABC = time-driven activity-based costing [26].

## Data Availability

Not applicable.

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
