# Peer review of "Economic Evaluations of Magnetic Resonance Image-Guided Radiotherapy (MRIgRT): A Systematic Review"

_ijerph, 2022, doi:10.3390/ijerph191710800_

Round 1
Reviewer 1 Report
In this manuscript titled " Economic Evaluations of Magnetic Resonance Image-guided radiotherapy (MRIgRT): a systematic review,". The authors undertake a comprehensive systematic review of the data on the economic effects of 16 different MRI-guided RT (MRIgRT). While this review emphasizes the promising future of 28 MRIgRT, it also emphasizes the need for further confirmation, particularly on the decrease of late toxicity, whose reduction is ex-29 pected to be the added value given by this technique.
Overall, this is a clear, concise, and well-written manuscript. However, one of the primary drawbacks of utilizing this method to conduct a systematic literature review in health economics is that it presumes familiarity with the field and the breadth of the existing studies. Both knowledge of the literature to identify potential search keywords and competence in creating methods are necessary for this procedure. Leaving out a crucial keyword may have the same effect as disregarding a considerable portion of existing literature. Further, any effort at a systematic review would be either too big or result in a local snapshot of the current literature. The authors should specify in every step of the systemic review what additional steps they have taken to include all the relevant studies and ensure the accuracy of the final results and how they handle every risk of bias.
Some issues need to be addressed by the authors.
Specific comments/inquiries are below:
Introduction:
More details about MRIgRT should be provided in the introduction.
Using this technology requires locating a considerable budget; as the authors mentioned, could the author put some estimations of those costs?
Methods
During the data collection process, it needs more details, is any form been used? If there was unreported information, did you contact the corresponding authors for that paper?
Results
From 735 articles, the author included only five articles. Could the authors explicitly list the reason why those articles were excluded?
Author Response
- In this manuscript titled " Economic Evaluations of Magnetic Resonance Image-guided radiotherapy (MRIgRT): a systematic review,". The authors undertake a comprehensive systematic review of the data on the economic effects of different MRI-guided RT (MRIgRT). While this review emphasizes the promising future of MRIgRT, it also emphasizes the need for further confirmation, particularly on the decrease of late toxicity, whose reduction is expected to be the added value given by this technique.
Overall, this is a clear, concise, and well-written manuscript. However, one of the primary drawbacks of utilizing this method to conduct a systematic literature review in health economics is that it presumes familiarity with the field and the breadth of the existing studies. Both knowledge of the literature to identify potential search keywords and competence in creating methods are necessary for this procedure. Leaving out a crucial keyword may have the same effect as disregarding a considerable portion of existing literature. Further, any effort at a systematic review would be either too big or result in a local snapshot of the current literature. The authors should specify in every step of the systemic review what additional steps they have taken to include all the relevant studies and ensure the accuracy of the final results and how they handle every risk of bias.
RESPONSE: We thank the reviewer for having appreciated our work and for the suggestions provided.
We are aware of the risk of a low search string sensitivity and have taken some measures to manage this risk. The working group is composed by researchers experienced in health technology assessment and in radiotherapy, also specifically in MRIgRT. We have better reported this in the manuscript (section 2.3). Furthermore, we are aware of the relevant literature on the subject. In particular, we consulted a recently published authoritative review: Hall, WA et al. Magnetic resonance linear accelerator technology and adaptive radiation therapy: An overview for clinicians. CA. Cancer J. Clin. 2022, 72, 34–56, doi:10.3322/caac.21707. To further increase the sensitivity of the searching phase we also performed a forward and backward citation tracking of the included studies, as reported in section 2.2.
Some issues need to be addressed by the authors.
Specific comments/inquiries are below:
Introduction:
- More details about MRIgRT should be provided in the introduction.
Response: we have provided additional details in the introduction.
- Using this technology requires locating a considerable budget; as the authors mentioned, could the author put some estimations of those costs?
RESPONSE: The logistical considerations associated with MRIgRT are well discussed in Hall, WA et al. 2022. We have added this reference. Regarding the costs, it is not easy to find a meaningful estimation as this is usually a confidential information within the negotiations between the company and the health center (for example, Parikh and colleagues – 2020 – report “Each of these estimates were taken from sales representatives as typical sales prices; actual sales prices vary and are subject to change”). Having an estimation of this cost is precisely one of the objectives of our work. In addition to the capital cost per patients, we have now added the purchase cost as reported by the authors of the included studies – see section 3.6 Model outcomes.
Methods
- During the data collection process, it needs more details, is any form been used? If there was unreported information, did you contact the corresponding authors for that paper?
Response: We have now reported that: a) data were schematically reported in specially defined forms; b) authors were contacted for unreported information. See section 2.6 Data Items.
Results
- From 735 articles, the author included only five articles. Could the authors explicitly list the reason why those articles were excluded?
Response: the reason for exclusion of the 728 studies based on their title and abstract is summarized in the initial part of the Section 3. Consistent with the PRISMA recommendations, the exclusion details are provided only for those works that passed the first screening phase (see Section 3 and Figure 1).

Reviewer 2 Report
#1. section 3.7 “In fact, the tools were designed for cost-effectiveness studies…We have therefore chosen not to undertake a formal quality assessment.” >> Please noted CHEERS 2022 [Pharmacoeconomics . 2022 Jun;40(6):601-609] stated “The CHEERS 2022 statement consists of a 28-item checklist… is intended to be used for any form of health economic evaluation and is primarily intended for researchers reporting economic evaluations for peer reviewed journals”, so, how about using CHEERS 2022 for a formal quality assessment?
#2. line 436 “marrow margins.” >> What did it mean?
Author Response
- Section 3.7 “In fact, the tools were designed for cost-effectiveness studies…We have therefore chosen not to undertake a formal quality assessment.” >> Please noted CHEERS 2022 [Pharmacoeconomics . 2022 Jun;40(6):601-609] stated “The CHEERS 2022 statement consists of a 28-item checklist… is intended to be used for any form of health economic evaluation and is primarily intended for researchers reporting economic evaluations for peer reviewed journals”, so, how about using CHEERS 2022 for a formal quality assessment?
Response: As clarified by the authors of CHEERS 2022, “The statement is not intended as a scoring tool or a tool to assess the appropriateness of methods”. Being a reporting standard similarly to the PRISMA statement adopted by us (see the PRISMA 2020 check list we produced for our manuscript), it is intended only for “researchers reporting economic evaluations for peer reviewed journals as well as the peer reviewers and editors assessing them for publication”. Several authors have adopted CHEERS for assessing the quality of economic studies in systematic reviews (see for example https://academic.oup.com/occmed/article/72/2/70/6472387, https://doi.org/10.1186/s12913-021-07025-8), but they introduced an arbitrary score.
Furthermore, as reported in the manuscript (Section 3.7), other tools could have been used but, given the great heterogeneity of the works identified, we have preferred to directly adopt a narrative approach in discussing the quality issues. Calculating a quality score would imply a comparison among the identified works which is not possible.
- Line 436 “marrow margins.” >> What did it mean?
Response: We thank the reviewer for having identified this misprint. We intended “narrow”, i.e., narrow margins between the planning target volume and the clinical target volume.

Round 2
Reviewer 1 Report
The author has addressed all my concerns. I have no further comments.
Good luck